# 3D-Printed PLA Molds for Natural Composites: Mechanical Properties of Green Wax-Based Composites

**DOI:** 10.3390/polym15112487

**Published:** 2023-05-28

**Authors:** Mihai Alin Pop, Mihaela Cosnita, Cătălin Croitoru, Sebastian Marian Zaharia, Simona Matei, Cosmin Spîrchez

**Affiliations:** 1Materials Science Department, Transilvania University of Brasov, 29 Eroilor Ave., 500484 Brasov, Romania; mihai.pop@unitbv.ro (M.A.P.); simona.matei@unitbv.ro (S.M.); 2Department of Product Design, Mechatronics and Environment, Transilvania University of Brasov, 29 Eroilor Ave., 500484 Brasov, Romania; 3Materials Engineering and Welding Department, Transilvania University of Brasov, 29 Eroilor Ave., 500484 Brasov, Romania; c.croitoru@unitbv.ro; 4Manufacturing Engineering Department, Faculty of Technological Engineering and Industrial Management, Transilvania University of Brasov, 29 Eroilor Ave., 500484 Brasov, Romania; zaharia_sebastian@unitbv.ro; 5Wood Processing and Furniture Design of Wood, Transilvania University of Brasov, 29 Eroilor Ave., 500484 Brasov, Romania; cosmin.spirchez@unitbv.ro

**Keywords:** green composites, beeswax, fir needles, recyclable paper, *Equisetum arvense*, mechanical properties

## Abstract

The first part of this paper is dedicated to obtaining 3D-printed molds using poly lactic acid (PLA) incorporating specific patterns, which have the potential to serve as the foundation for sound-absorbing panels for various industries and aviation. The molding production process was utilized to create all-natural environmentally friendly composites. These composites mainly comprise paper, beeswax, and fir resin, including automotive function as the matrices and binders. In addition, fillers, such as fir needles, rice flour, and *Equisetum arvense* (horsetail) powder, were added in varying amounts to achieve the desired properties. The mechanical properties of the resulting green composites, including impact and compressive strength, as well as maximum bending force value, were evaluated. The morphology and internal structure of the fractured samples were analyzed using scanning electron microscopy (SEM) and an optical microscopy. The highest impact strength was measured for the composites with beeswax, fir needles, recyclable paper, and beeswax fir resin and recyclable paper, 19.42 and 19.32 kJ/m^2^, respectively, while the highest compressive strength was 4 MPa for the beeswax and horsetail-based green composite. Natural-material-based composites exhibited 60% higher mechanical performance compared to similar commercial products used in the automotive industry.

## 1. Introduction

The circular (bioresources) economy is considered one of the most sustainable potential solutions to the global challenges of energy crises and environmental pollution. The research community has approached this issue from two main paths: recycling waste to develop novel materials and using natural resources to create new materials. One area of significant attention for researchers is the development of fiber-reinforced composite materials, which have the potential to be used in (semi) structural applications [1]. In the last decade, great attention has been paid by the research community to the developments of fiber (carbon, graphite, graphene, aramid, kevlar, natural, and wood fibers)—polymer (thermoplastic or thermosetting matrices) composites. Fiber-reinforced composite materials (FRPs) are widely used in modern products due to their exceptional combination of properties, including high specific strength and stiffness, low weight, durability, and resistance to creep and fatigue. Due to the special properties in relation to weight, composite materials have found a safe place in top industries, such as aerospace and aeronautics (applications for propellers of torpedoes, submarines, etc.) [2], in the automotive industry (components of transmission, brake discs, engine components) [2,3,4,5,6,7], in electronics and electrotechnics (manufacture of packages and layers of integrated circuit-plate supports on which electronic circuits are mounted to capacitors, electronic transducers or electronic filters) [8], and so on. However, achieving adequate interfacial strength between the filler and matrix often requires the use of toxic or expensive compatibility agents. Additionally, when synthetic fibers are used, the manufacturing process can become energy intensive due to the high-temperature curing required for the fiber–polymer blend [9,10].

Extensive studies in the specialized literature indicate that many composite materials with natural reinforcements use polymeric matrices that exhibit good or even very good physical–mechanical properties [11,12]. However, the advantage of these materials is diminished by the polluting factor over their lifetime, as well as the high costs and complex technological/chemical processes required for their manufacturing and the limited recycling options available at the end of their product lifecycle [13,14,15]. The environmental sustainability and cost effectiveness has pushed researchers’ focus to be moved from synthetic FRP composites to more environmentally friendly fillers. Natural plant fibers have emerged as a suitable alternative to synthetic fibers due to their lightweight nature, renewability, sustainability, eco-friendliness, carbon neutrality, low cost, biodegradability, and availability. Moreover, these fibers provide good insulation, toughness, vibration damping, flexibility, and high specific strength and modulus [16,17]. Moreover, legislative pressures on the industrial sector have led to an increased demand for environmentally friendly materials. As a result, natural fibers have gained traction as a commonly used alternative to synthetic fibers in composite materials. Flax, hemp, jute, kenaf, banana, pineapple, and sisal are amongst the most commonly used natural plant fibers as reinforcements in composite materials [1,9,11,17,18,19,20]. Compared to synthetic carbon and glass fibers, natural plant fibers are less expensive, lighter, biodegradable, easy to produce, and environmentally friendly. Researchers are now exploring the use of natural fibers for higher-load applications by incorporating ceramic fillers and synthetic fibers as reinforcements. This shift towards biologically sourced and recyclable materials is driven by the growing demand for sustainable and environmentally friendly materials in order to diminish the burden on the environment [21,22]. A study on ramie-reinforced PLA composites was reported, with a positive impact on the mechanical properties (tensile strength) of ramie/PLA composites [17,19]. Farid et al. [23] investigated the use of natural fibers, including kenaf, jute, waste cotton, and flax, in blends with PP and polyester binders for sound-absorbing floor coverings. They found that the sound absorption coefficient increased with the frequency of sound [24]. This research highlights the potential of natural fibers in creating sustainable products. In this context, the assimilation of pine needles (PNs) through biotechnology can offer a valuable opportunity to enhance the circular bio-economy. Converting PNs into biomaterials and bio-energy can help reduce the reliance on petroleum products and promote a healthier environment by maintaining the energy–environment network [22,23,24,25,26,27,28,29,30,31,32,33,34,35,36,37].

Composite material development by using natural fibers promotes reductions in greenhouse gas emission, biodegradability, job availability increase, energy saving, easy decomposition, lightweight, high specific mechanical properties, less tool wear during processing, and less density, lowering the product cost. However, there are some shortcomings when using natural fiber composites due to their hydrophilic nature and the potential of moisture absorption, debonding the fillers from the matrix [30,31,32,33,34,38]. The CO_2_ release during when obtaining natural fiber composites is negligible as compared to synthetic fiber [35]. The composite structure comprises biodegradable natural fibers, which have found significant applications as sound-proofing material for automobile components. Composite laminates with bamboo, cotton, and flax fibers with PLA fibers showed bending stiffness of 2.5 GPa, but the impact strength, another mechanical property required in such applications, was been reported [36,37].

Beeswax is derived from honey combs and is shown as a complex mixture of several chemical compounds mostly used in the food industry (glazing, carrier of additives in food, and texturizing agent in chewing gum production) [38]. Praveen B. et al. [38] blended cellulose triacetate (CTA) microspheres and beeswax in order to evaluate the controlled release of antidiabetic nateglinide [39]. Pine needles and natural materials were used to obtain valuable products, metabolites, and bio-energy [23,24,28]. Converting natural materials into valuable products not only promotes a sustainable energy–environment network but also reduces the use of conventional petroleum products, resulting in a positive impact on the environment and human health [32,34]. While technology has advanced significantly, virtual information storage cannot completely replace readable paper supports, leading to the continued generation of paper waste.

To the best of our knowledge, there are no previous studies on the fabrication of green composites by pouring natural material blends based on beeswax, fir resin, fir needles, and *Equisetum arvense* onto recyclable paper.

This work presents a novel approach for the development of eco-friendly composites, consisting of two stages. Firstly, specific molds were 3D-printed using PLA to mold the natural composite receipts based on recyclable paper, beeswax, fir resin, fir needles, rice, and *Equisetum arvense*. In the second stage, eco-composites were prepared through mold casting (pouring) several natural blends of beeswax, fir resin, with or without fir needle, rice, or *Equisetum* onto recyclable paper. This study aims to contribute to the development of sustainable materials and to reduce waste generation from paper consumption, while also exploring the potential of natural resources for composite material production.

The major advantage of these composite materials is that both the matrix and reinforcement are natural, biodegradable, and recyclable.

This study is part of a larger research program aimed at assessing the feasibility of using the obtained composites for various industrial applications, such as sound insulation, thermal-insulating panels, or shock absorbers. In this particular study, we will focus specifically on the mechanical properties of the composites. By examining the mechanical properties, we can better understand how these materials can be optimized to provide the necessary insulation and damping and to gain a more specific insight into their potential application domains.

## 2. Experimental Section

### 2.1. 3D-Printing Molds for Green Composite Development

The research employs a variety of natural materials, such as polymers, resins, waxes, and biomass waste, to produce composite materials with good sound-absorbing properties. These materials, which include beeswax, fir resin, ground fir needles, horsetail (*Equiseti herba*), rice flour, and paper pulp, offer a high degree of recyclability, low cost, and minimal environmental impact.

Natural wax is an excellent candidate for use as an organic matrix in sound-absorbing materials, due to its high content of saturated fatty acids, n-alkanes, and long-chain alcohols. Similarly, fir resin, with its abundance of resin acids, provides an ideal matrix material for these composites. Not only do these materials possess excellent sound-dissipating properties due to their amorphous nature but they are also easy to process, have low softening points, and act as effective binders for filler materials. This combination of properties makes them an attractive option for creating effective sound-absorbing materials at a low cost.

In the pre-testing phase, it was determined that the most suitable filler material for the composite was in ground form. This decision was based on the groundings’ relatively large specific surface area, which enables them to bond more effectively with the matrix material and also helps to create a structure that is as loose as possible. This loose structure is particularly beneficial for sound absorption, as it allows sound waves to be effectively dissipated within the material.

*Equisetum* and fir needles are not only rich in lignocellulosic material but also in phenolic compounds, resin acids, and silicon dioxide (in the case of horsetail), which contribute to the strengthening and thermal stabilizing effect of wax and/or resin. Rice flour, on the other hand, contains starch, waxes, and proteins that are compatible with the chosen matrices, while milled paper (recycled cellulose) has a high specific surface area and absorbency compared to wax and resin. These unique properties of the natural materials make them excellent candidates for the development of eco-friendly composites with enhanced mechanical and thermal properties.

Thus, the following eco-composites were preliminarily analyzed from the point of view of SHORE D hardness:(a)Beeswax (Ca) and ground horsetail (Cc)—a total of 14 composite materials were obtained. Percent of ground horsetail between 0 and 39.4%.(b)Beeswax (Ca) and recycled paper (Hr)—a total of 18 composite materials were obtained. Percent of recycled paper was between 0 and 46%.(c)Beeswax (Ca) and ground rice (Or)—a total of 10 composite materials were obtained. Percent of ground rice was between 0 and 86.5%.(d)Beeswax (Ca) and ground fir needles (Abr)—a total of 10 composite materials were obtained. Percent of ground fir was between 0 and 86.5%.

A common/household grinding machine was used for grinding and bringing it to the desired size, and to determine the granulation size, a system equipped with mesh sieves between 0.036 mm and 0.5 mm diameter was used.

For each material in powder form, the amount was gradually increased step by step with 0.25 g and then calculated as a percentage of the entire composite.

First of all, all the elements (recycled cellulose, *Equiseti herba*, rice flour, and paper pulp) were chopped to the desired granulation, weighed, and prepared according to the preset recipes for the next step, namely mixing with beeswax.

The second step was to create the composites: beeswax was melted (up to a temperature of 100 °C) and mixed with various reinforcements (presented above) in different proportions until the reinforcer/hardener was maximally moistened.

Last step was to pour the composite while it was still in a pasty state in the mold.

The combinations that produced the best results were identified, and we decided to use these combinations as a basis for developing new composite materials, as follows:Recyclable paper 100% denoted as S1 (as base for comparison with new composites)Beeswax (50%) + fir resin (50%) denoted as S2Beeswax (62.5%) + horsetail (37.5%) denoted as S3Beeswax (55.56%) + recycled paper (44.44%) denoted as S4Beeswax (45.5%) + milled rice (54.5%) denoted as S5Beeswax (61.5%) + milled fir needles (38.5%) denoted as S6Beeswax (31.25%) + fir resin (31.25%) + horsetail (37.5%) denoted as S7Beeswax (27.78%) + fir resin (27.78%) + recycled paper (44.44%) denoted as S8

To assess the mechanical properties of the natural composite materials, including their flexural strength, impact resistance, and compression resistance, specific molds were designed using SolidWorks 2016 software. These molds were specifically tailored to the unique characteristics of the natural composite materials under study. The molds were then created using 3D-printing technology, utilizing the Fused Filament Fabrication (FFF) technique to produce positive molds.

Silicone rubber was subsequently cast into the positive molds, creating negative molds that were then used to shape the natural blends into their final form. These natural blends were poured into the negative molds, which were designed to be made from recyclable paper, enabling a sustainable and environmentally friendly production process. By utilizing this process, precise and accurate measurements of the natural composite materials’ mechanical properties were obtained, providing valuable insights into their potential applications.

International Standard ISO 14125—Plastics, Subcommittee SC 13, Composites and reinforcement fibers—was the basis for testing the samples but not the only one; ISO 178 or ASTM D 790 was consulted, for example.

The photo images taken from the 3D-printed molds are presented in Figure 1.

This method of obtaining the molds for testing the mechanical properties of the samples was chosen due to the particularities of these eco-composites: precise, simple, and efficient.

### 2.2. Materials and Green Composite Preparation

#### 2.2.1. Materials

The green composite materials were developed using natural materials, with beeswax and fir resin serving as the matrix and fir needles, rice, and *Equisetum arvense* as the reinforcing materials. These materials were selected for their biodegradability, recyclability, and sustainable sourcing. Specifically, the fir resin and fir needles were collected during trips to coniferous forests in Brasov, Romania, from fallen branches on the ground. The natural beeswax was purchased from Apsirom SRL Vaslui, Romania, while the allimentary (grocery-grade) rice and *Equisetum arvense* were obtained from a local shop in Brasov, Romania.

The raw materials used for developing green composite materials are presented in Figure 2.

All these natural materials used for green composite preparation were dried and ground before their blending. After grinding the fir needles, the rice and *Equisetum arvense* grain diameter was under 0.5 mm.

#### 2.2.2. Obtaining Composites

The following steps were taken to prepare the green composite materials:(1)The base natural materials (beeswax and/or fir resin) were melted at a temperature of approximately 80–100 °C.(2)The reinforcing natural filler powders (fir needles, rice, *Equisetum arvense*) were added in the proportions specified above.(3)The natural material blend was mixed thoroughly.(4)The blend was poured into silicone rubber molds, as shown in Figure 1, over the recyclable paper.

The recyclable paper was prepared according to the process outlined in Figure 3.

The green composites obtained after pouring the natural material blends onto the recyclable paper can be seen in Figure 4.

All the natural composition blends were poured onto recyclable paper in the PLA molds.

### 2.3. Characterization Techniques

#### 2.3.1. Mechanical Tests

Bending and compression strength and Young’s modulus (E) for the composite structures (sandwich structures and wing sections) were performed on the WDW-150S universal testing machine with a constant crosshead speed of 10 mm/min for all test (Jinan Testing Equipment IE Corporation, Jinan, China). The impact (resilience) strength of the green composites obtained was measured on a Galdabini Impact 25 equipment, Cardano al Campo, Italia, with 25 kJ maximum energy. Compression and bending tests were performed with a speed of 10 mm/s. The impact energy of the impact hammer was 5.5 J. Five samples were tested for each mechanical test property, and the average values are reported.

#### 2.3.2. Surface Morphology Analysis

Micrographs were obtained by using a scanning electron microscope (SEM), Hitachi, Japan, S3400N, type II, and the images were taken from the impact-fractured composite surface.

#### 2.3.3. Optical Microscope

Images of the fractured composite surfaces were taken using an optical microscope type Leica (Arnhem, The Netherlands), Emspira 3 model.

## 3. Results and Disscusion

### 3.1. Mechanical Tests

The mechanical, thermal, and durability properties of a composite are largely determined by the interfacial adhesion zone in the composite system. The strength of this zone depends on several factors, including the physical–chemical properties of each composite component, their size and shape, mass ratio, dispersion grade of fillers into the matrix composite, preparation technique, and technological parameters used for composite preparation. In this study, we evaluated the mechanical properties of the green composites in terms of impact, compression, and bending maximum force, as shown in Figure 5, Figure 6 and Figure 7.

Figure 5 shows the variation in impact resistance (resilience) for the natural composites. It is worth noting that the addition of the melted beeswax and fir resin positively influences the impact strength of the resulting composites (S2 and S3) compared to S1 (recyclable paper only). The addition of *Equisetum arvense* powder (S4) and ground rice (S5) negatively impacted the resilience value of the beeswax-based samples (S2), likely due to their higher modulus of elasticity, which includes silica-based compounds and unplastified starch. Conversely, the addition of fir needle powders (S6) and fir resin (S8) led to an increase in shock resistance or resilience. The best resilience value was achieved with the S6 composite, which was based on beeswax and fir resin. Additionally, improvements in impact strength were recorded for S2, S6, S7, and S8, with the highest values found in S6 and S8 at 19.42 and 19.32 kJ/m^2^, respectively. These results can be assigned to the mechanical strength provided by fir resin, as well as the viscoelastic properties of the cellulosic fibers, as noted by Jakob et al. [40]. It is widely recognized that cellulosic fibers significantly contribute to the mechanical enhancement of fiber-based composites [40,41,42,43,44]. These findings are consistent with a study conducted by Butnaru et al. [44], which demonstrated the superior thermal and mechanical properties of fir needles compared to fir cone and bark. Furthermore, in addition to their remarkable mechanical and thermal strength, fir needles also exhibit antioxidant properties, which are of great importance in the development of natural composite materials [35].

Figure 6 shows the compression strength of the natural blends poured onto recyclable paper. Sample S2, based on beeswax poured onto recyclable paper, exhibited a 66.6% increase in compression strength, while sample S3 showed a remarkable increase, exceeding 100%, from 1.8 MPa recorded for S1 (recyclable paper) to 4 MPa recorded for S3. This significant increase in compression strength can be attributed to the insertion of melted fir resin molecules through the paper fibers, as well as the formation of an ordered shell that covers the paper core, as observed in the next section using light microscopy. The hardened fir resin in S3 effectively reinforced the paper fibers, while the shell provided additional support, resulting in a remarkable increase in compression strength.

Ground rice addition caused a slight decrease in compressive strength for the S5 sample, while the addition of fir needle powder caused a slight increase in compressive strength for the S6 sample. The sample based on fir resin (S3) showed the best value for compressive strength. However, the addition of *Equisetum* powder and beeswax (S7) and beeswax alone (S8) had a strong negative effect on the compressive strength of sample S3.

In Figure 7, the maximum bending force of the composites is presented. The addition of ground fir needles to the S2 sample caused a decrease in the maximum bending force for the S6 composite. On the other hand, the addition of *Equisetum* powders (S4), ground rice (S5), and fir resin with *Equisetum* powders (S6) led to an increase in the maximum bending force for sample S2. Among all the samples, composite S5 based on beeswax and ground rice demonstrated the best value for maximum bending strength. This can be attributed to the swelling of the rice fibers, as can be observed in the inset surface fracture image of S5 in Figure 7. The rice fibers covered the paper core, resulting in higher bending force compared to other samples.

In this study, we compared the mechanical properties of the green composites, which were considered as the core materials for the final green products with those of similar commercial products from the automotive sector (see Table 1). We mechanically tested the commercial core product used as a sound-absorbing panel and found that its properties were inferior to those of the core material composites developed in this research. These results indicate that the green composites have the potential to be a superior alternative to the commercial products in terms of mechanical strength and could be applied in various fields, including automotive and sound-absorbing panels.

The results obtained from the mechanical tests on the proposed composites indicate that improvements and a different approach are necessary to further enhance these properties without sacrificing their advantages, such as being natural, ecological, recyclable, and sustainable. However, the improved physical–mechanical properties make these composites suitable for various applications beyond the automotive industry, including aeronautics and other fields where sound-absorbing panels are needed.

It was also observed that high mechanical properties are not always required for materials used in the automotive industry, particularly for the lining of the engine hood and linear luggage, where good sound-absorbing properties are more important. Overall, the use of natural, sustainable materials in these applications can provide a more environmentally friendly solution while maintaining or even improving performance.

### 3.2. Light Microscopy

To observe the effect of the natural material blend incorporation poured onto the recyclable paper matrix, optical images of the fractured green composites’ surfaces were analyzed using light microscopy. Figure 8 displays representative images of all samples, where the lowest incorporation of natural material onto the recyclable paper matrix corresponds to sample S3, which was poured with fir resin. The rapid toughening of the fir resin forms a stiff shell that encapsulates the paper matrix, resulting in the highest compression strength of this sample compared to the others, as previously shown in the compression test with a recorded value of 4.6 MPa.

The observations from the light microscopy images show that the natural material blends were able to wet the paper core effectively. The dispersion of natural blends was remarkable for samples S2, S6, S7, and S8, with the blends encapsulating the paper core fibers, thereby adding mechanical strength to the resulting green composites. These findings confirm the results from the mechanical tests, particularly the high impact strength values of 19.16, 19.42, and 19.32 kJ/m^2^ for the green composites denoted as S2, S6, and S8, respectively.

### 3.3. Surface Morphlogy and Internal Structure

Scanning electron microscopy was performed to investigate the surface morphology and interface structure of the impact-fractured composite samples with good mechanical strength (S2, S6, S7, and S8), as shown in Figure 9. The SEM images reveal a low-rugosity surface, indicating strong bonding between the composite components and reflecting good interface strength.

The SEM images obtained from the fractured S2, S6, and S7 samples demonstrate a uniform morphology, indicating a high dispersion of the melted natural materials (beeswax, fir resin with/without fir needles, and *Equisetum* fibers) throughout the internal structure of the paper fiber cores. This behavior, particularly in the case of the S2 sample, confirms its mechanical performance, as the mechanical tests showed an approximately doubled impact strength compared to the recyclable paper-only reference. The interface failure is observed as a matrix failure, with tearing and shearing of the matrix visible. The images in Figure 9 also show that even after composite fracture, the paper core fibers are still covered by the melted natural material blend.

## 4. Conclusions

This paper presents the steps involved in obtaining 3D-printed PLA molds for natural composites made of recyclable paper, beeswax, fir resin, needles, rice, and *Equisetum arvense*, and it assesses the mechanical properties of the green composites;The highest impact strengths, provided mainly by fir resin and cellulose, were attributed to the samples with beeswax and fir needles S6 and beeswax fir resin and recyclable paper S8, 19.42 and 19.32 kJ/m^2^, respectively, while the highest compressive strength was 4 MPa for the S3 sample. The rice fibers positively influenced the maximum bending for the S5 sample;The superiority of the green composites’ mechanical properties was proved, considering the core materials for the final green products over similar commercial products from the automotive sector;The green composites exhibited over 60% higher mechanical properties compared to similar products from the market used as sound-absorbing core materials in the automotive industry;The physico-mechanical properties were directly influenced by the wetting degree of the powders for the beeswax and the fir resin and through the interaction between the complementary chemical groups from the fillers and the matrices (possibly silicates, phenolic carboxylic acids, resinic acids, and so on);Further research is ongoing to determine the thermal and sound-absorbing properties of these composites, as well as finding new solutions to enhance their physical–mechanical properties. One possible solution is combining the benefits of eco-friendly filament printing, such as polylactic acid, with the sound-absorbing properties of these new materials.

## Figures and Tables

**Figure 1 polymers-15-02487-f001:**
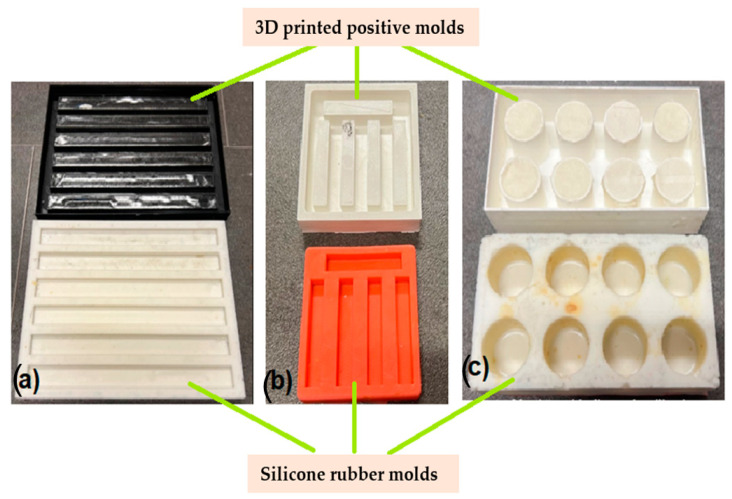
3D-printed positive and negative molds for mechanical testing of the green composites: (**a**) for bending; (**b**) for impact; (**c**) for compression.

**Figure 2 polymers-15-02487-f002:**
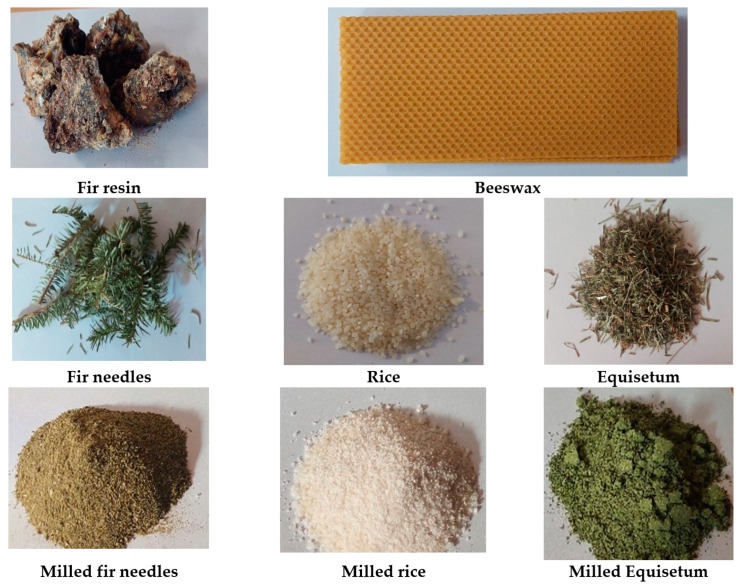
Raw natural materials used for the green composites.

**Figure 3 polymers-15-02487-f003:**
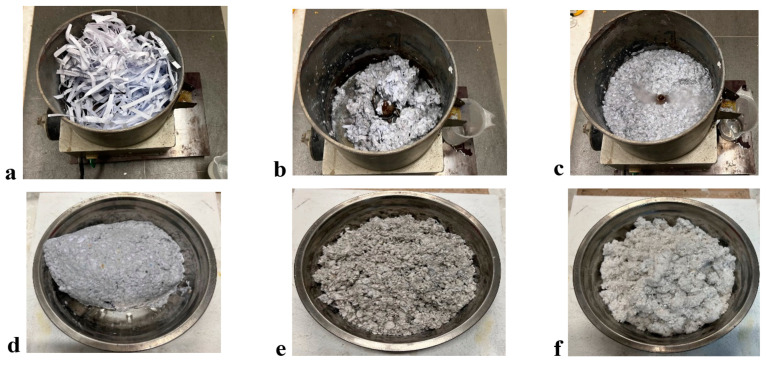
Steps in preparing the recyclable paper: (**a**) shredded paper; (**b**) water addition; (**c**) homogenizing with a blender; (**d**) paste sieving; (**e**) drying the paper paste; (**f**) grinding recyclable paper.

**Figure 4 polymers-15-02487-f004:**
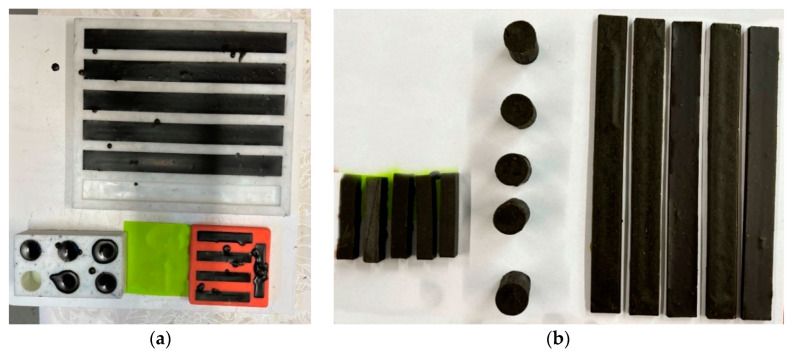
Green composites prepared for mechanical tests; (**a**) casting in the silicon rubber molds; (**b**) samples removed from the molds.

**Figure 5 polymers-15-02487-f005:**
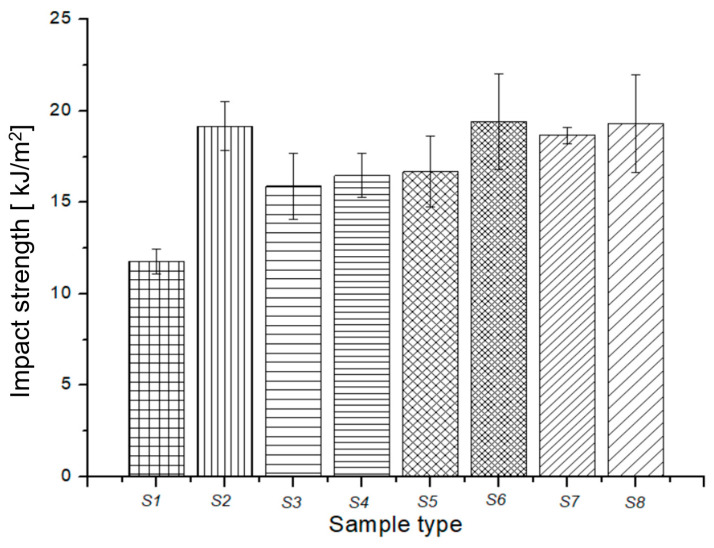
Impact strength variation for green composites.

**Figure 6 polymers-15-02487-f006:**
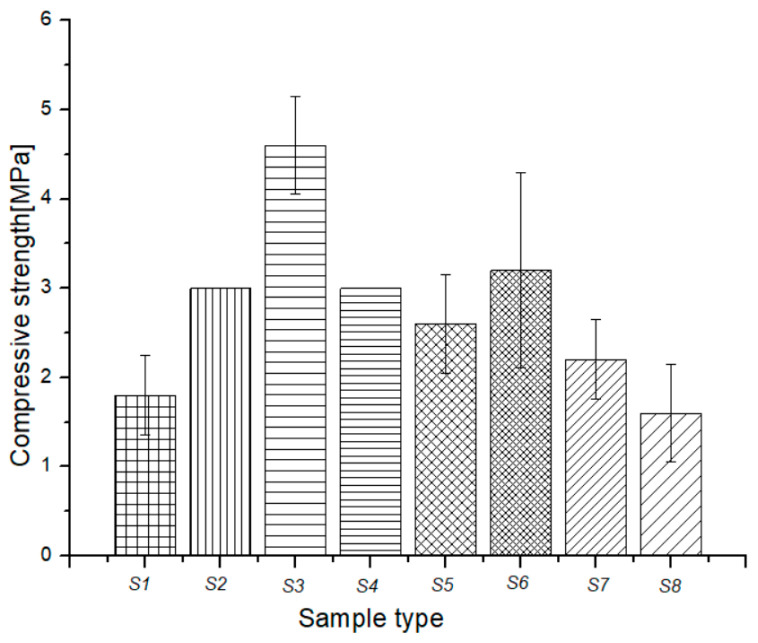
Compressive strength variation for green composites.

**Figure 7 polymers-15-02487-f007:**
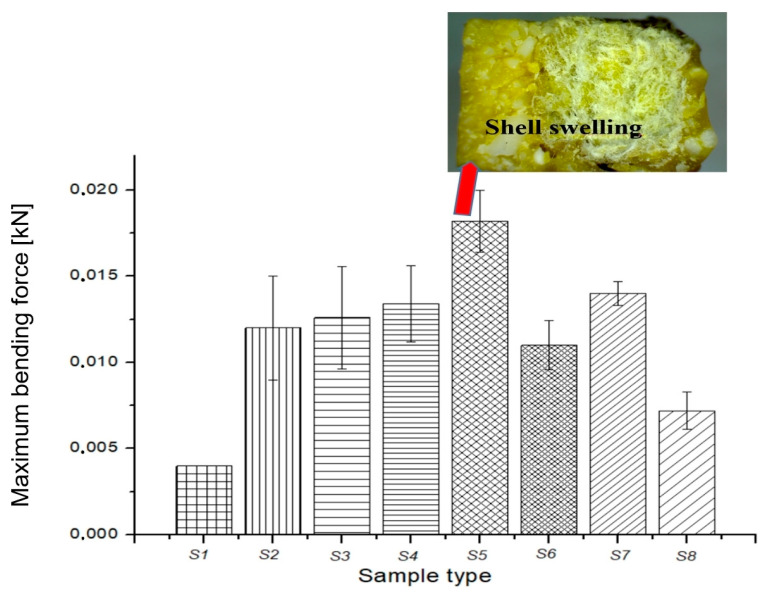
Maximum bending force variation for composites.

**Figure 8 polymers-15-02487-f008:**
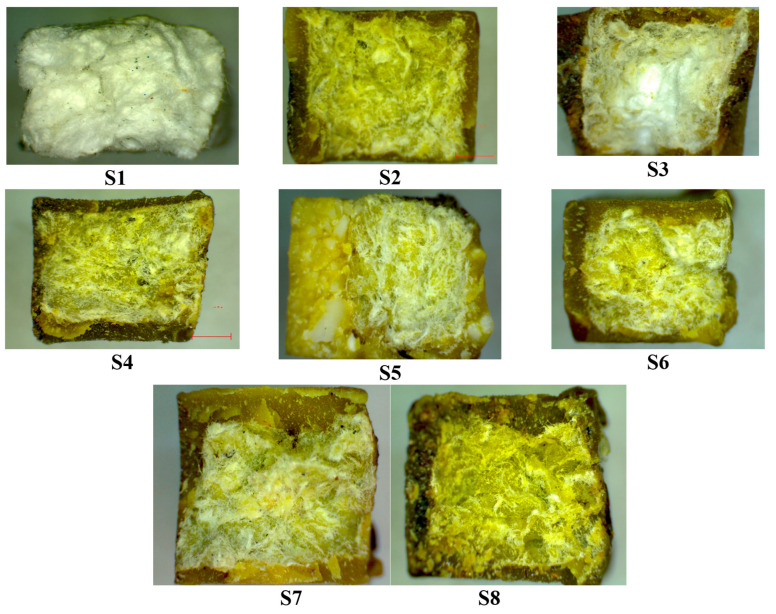
Fractured surfaces’ microscope images of the green composites (30× magnification).

**Figure 9 polymers-15-02487-f009:**
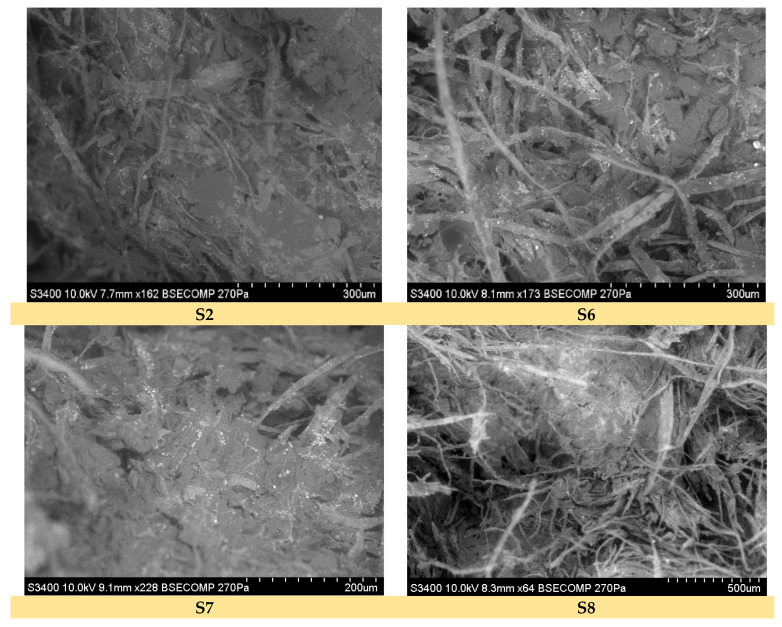
SEM images of fractured green composites’ surfaces.

**Table 1 polymers-15-02487-t001:** Mechanical properties of the green composites compared to similar commercial products used in the automotive industry.

Mechanical Property/Commercial Automotive Sample/Green Composites	Impact Strength [kJ/m^2^]	Compressive Strength [MPa]	Bending Force [N]
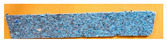 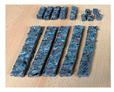	11.41	<1	<3
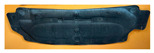 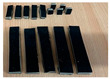	13.26	<1	<3
**Green composites’ mechanical properties**			
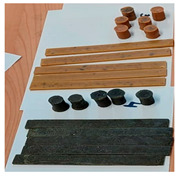	19.42	4.60	18.20

## Data Availability

The data presented in this study are available on request from the corresponding author.

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
