# Peer review of "3D-Printed PLA Molds for Natural Composites: Mechanical Properties of Green Wax-Based Composites"

_polymers, 2023, doi:10.3390/polym15112487_

Round 1

Reviewer 1 Report

The paper presents the application of wax in the production of green composites in terms of the promotion of the circular economy. The introduction of the paper is well-written and presents the overall background of the problem considered. The methodology of the research are written clearly. The results are shown in the proper way using both tables and figures. The conclusions presents main results of the research performed. The references are selected properly and their number is adequate for scientific papers.

Generally, the paper is interesting and well-written, as well as, considers the very important issue concerning the application of waste materials in the production of polymers which is consistent with the recycling, as well as, the circular economy idea.

Some remarks before the final acceptance of the paper:

Fiber reinforced composite materials (FRP) are widely used in modern products…” – give some examples of such products, please.

“Extensive studies in the specialized literature indicate that many composite materials with natural reinforcements use polymeric matrices that exhibit good or even very good physical-mechanical properties” – references are needed.

Conclusions should be written in points – they will be more readable for readers.

The English should be checked by native speaker – there are some mistakes.

English will be checked by native speaker - there are some mistakes.

Author Response

Dear Reviewer

Thank you very much for all your comments and suggestions.

Attached there are our responses.

Reviewer 2 Report

As the Polymers journal the topic of the work at hand would appear to be an appropriate one, in particular paying attention to the research.

The article briefly presents composites based on natural materials. The course of the research and methods has been presented correctly.

The abstract is a little bit confuse and missis some information like more results and conclusions.

 The Introduction section quite briefly refers to the content of the article, of course the authors pay attention to the key theses from the area of literature analysis, but this section should be reduce by a general thematic introduction.

It would be reasonable for the reader to introduce analysis of the properties of such composites.

most important notes:

-the research methodology should be described in detail, including the preparation of materials for microscopy,

- fig 8 no magnification information, there is no description of the preparations,

-no detailed description of the preparation of composites - machinery, equipment, mixing parameters, etc.

-why only five samples were tested, please refer to the standards for testing mechanical properties,

- measurement parameters are not described

- the test results should be analyzed and the reasons for the changes in properties obtained should be indicated,

- the text needs editorial correction in accordance with the requirements of the journal and the arrangement of photos in the figures needs to be improved; font, references (Fig 1, 2), etc.

The conclusions do not refer to the work, but to the description of what the work presents. It is recommended to conduct a deeper discussion and refer to the results in the conclusions, also critically presenting the advantages and disadvantages of the method - which does not seem to be difficult when reading the paper.

Minor editing of English language required

Author Response

Dear Reviewer,

Thank you very much for your time spent going through our paper. Your comments and suggestions helped us to provide a better version of our paper.

Our responses to your comments could be found in the attach.

Round 2

Reviewer 2 Report

The article has been corrected and may be published in its present form

Minor editing of English language required